# Text-to-Model: Text-Conditioned Neural Network Diffusion for Train-Once-for-All Personalization

## Abstract

Generative artificial intelligence (GenAI) has made significant progress in understanding world knowledge and generating content from human languages across various modalities, like text-to-text large language models, text-to-image stable diffusion, and text-to-video Sora. While in this paper, we investigate the capability of GenAI for *text-to-model* generation, to see whether GenAI can comprehend hyper-level knowledge embedded within AI itself parameters. Specifically, we study a practical scenario termed train-once-for-all personalization, aiming to generate personalized models for diverse end-users and tasks using text prompts. Inspired by the recent emergence of neural network diffusion, we present `Tina`, a text-conditioned neural network diffusion for train-once-for-all personalization. `Tina` leverages a diffusion transformer model conditioned on task descriptions embedded using a CLIP model. Despite the astronomical number of potential personalized tasks (e.g., $1.73 \times 10^{13}$), by our design, `Tina` demonstrates remarkable in-distribution and out-of-distribution generalization even trained on small datasets ($\sim 1000$). We further verify whether and how `Tina` understands world knowledge by analyzing its capabilities under zero-shot/few-shot image prompts, different numbers of personalized classes, prompts of natural language descriptions, and predicting unseen entities.

## 1 Introduction

Generative artificial intelligence (GenAI) has been flourishing in different aspects of human life, and people can simply generate content from natural language text prompts [1, 2, 3, 4]. Large language models [1, 5], like GPT-4, have especially shown emergent intelligence [6] in the knowledge of language through *text-to-text* transformation [7, 8, 1, 5]. Besides, recent progress in *text-to-image* (e.g., stable diffusion) [9, 4, 2, 10] and *text-to-video* (e.g., Sora) [3, 11] diffusion models has shown the great power of AI in understanding the physical world and generating high-quality images and videos that are virtually indistinguishable from reality [12, 3]. The text-prompted GenAI maps the human languages' semantics to the world

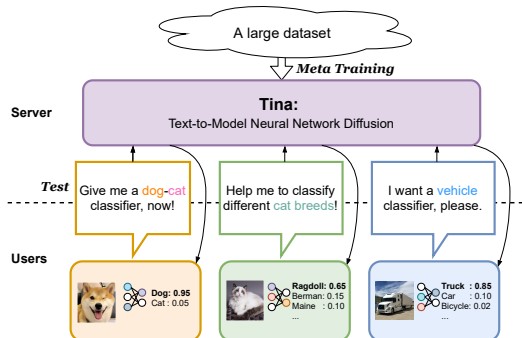

Figure 1: **Demonstration of train-once-for-all personalization scenario.** Users have text descriptions of the desired personalized models.

knowledge in different forms in language and vision. One step further, in this paper, we propose and study whether the GenAI can understand hyper-level knowledge—the knowledge inherently resides in the AI itself models' parameters. Specifically, we study **text-to-model** generation; akin to text-to-text, text-to-image, and text-to-video, text-to-model targets whether the GenAI models can

directly generate the model parameters given the human's text prompts to meet the personalization demand of diverse end users.

We focus on a practical scenario called train-once-for-all personalization [13], which means that the generic model is trained just once and can later be customized into a condensed model on the fly for different end-users and requests, given their task descriptions. For example, the CIFAR-100 dataset [14] contains 100 classes, but an end user may just need a personalized model with a certain 10 classes according to a specific scenario (e.g., classifying items in the kitchen). In other words, train-once-for-all personalization targets that train the model once and customize the model to be well performed in a sub-distribution when deployed, and an example is in Figure 1. But there are tremendous sub-distributions, for the CIFAR-100 example, the number of personalized 10-way tasks is $\binom{100}{10} = 1.73 \times 10^{13}$, even not taking permutations into consideration, so it is challenging for the GenAI model to generalize. Inspired by recent progress in neural network diffusion [15, 16], we propose Tina, a **T**ext-Conditioned **N**eural Network **D**iffusion for Train-Once-for-**A**ll Personalization. Tina is trained on model parameters with the models' task descriptions, and it can be generalized to unseen tasks, or even unseen classes (entities), given the text prompts.

In Tina, a CLIP model [17] is used to embed the users' task descriptions into the diffusion model as the conditions. The diffusion model of Tina is the diffusion transformer (DiT) [12] that is shown to have high expressive power under scaling law in the fields of image [12] and video generation [3]. We demonstrate that DiT's scaling law applies to model parameter generation as well: increasing the number of parameters and data sizes enhances the model's capability to generalize across more challenging tasks that involve scaling the dimension of generated models. However, it is surprising to find that even though the number of personalized tasks is astronomical (e.g., $1.73 \times 10^{13}$ for 10-way tasks), by our designs, Tina can generalize on extremely small datasets ($\sim 1000$ data points) and support different lengths of classification tasks (5-way or 8-way tasks, etc.) in training once. Our analysis shows that Tina can reach both in-distribution and out-of-distribution personalization of generated models. Thanks to the vision-language alignment of CLIP, Tina can also take images as prompts and generalize under few-shot or even zero-shot settings. We also verify whether Tina understands world knowledge by testing its abilities under prompts of natural language descriptions and predicting unseen entities. Our contributions are as follows:

- We explore the potential of GenAI in generating personalized models followed by users' text prompts, i.e., text-to-model generation. We open more applications of neural network diffusion; to the best of our knowledge, it is the first paper that takes the text prompts as conditions for neural network diffusion.

- We propose Tina, a well-performed text-conditioned neural network diffusion framework for train-once-for-all personalization. Tina can generalize on unseen tasks and entities even given small model datasets.

- In addition, we analyze the abilities and the boundaries of Tina and gain insights about whether and how it generalizes and understands world knowledge.

## 2 Methodology

### 2.1 Problem Setup

#### 2.1.1 Definition of Setup

Following [13], we consider image classification for train-once-for-all personalization due to the natural personalization requirements of image classification. We note that our method is not limited to classification tasks and can be extended to other tasks for personalization. Define a task $k$ as classification over a subset of classes $\mathcal{Y}_k \subset \mathcal{Y}$. The goal of personalization is to learn a neural network predictor $f_{\theta_k} : \mathcal{X} \mapsto \mathcal{Y}_k$, parameterized by $\theta_k$. To handle many tasks at the same time, we further assume we have the task description natural text $t_k$ for $\mathcal{Y}_k$, and it is generally the description of the classes and styles of $\mathcal{Y}_k$. We want to build a neural network generator $G(t_k)$ where given $t_k$, it will output the model parameters $\theta_k$. Specifically, consider using a large-scale dataset with many classes covering $\mathcal{Y}$ to learn the personalized-friendly function $f_{\theta_k} = G_\phi(t_k)$ parameterized by $\phi$. $G_\phi$ is learned on the large dataset to generate any personalized model directly from the task descriptions, and the setup is called train-once-for-all personalization [13]. Train-once-for-all personalization has wide applications in a server-user system, where the model generator $G_\phi$ is learned on the server for personalized cloud services to many future users. We refer to [13] for more detailed advantages and usages of train-once-for-all personalization.

### 2.1.2 Strong Baselines: Classifier Selection and TAPER

**Classifier Selection.** For a generic network $f_\theta$, we consider that it consists of a feature extractor parameterized by $\psi$ with a linear classifier $\mathbf{w} = [\mathbf{w}^{(1)}, \ldots, \mathbf{w}^{(|\mathcal{Y}|)}]$ of $|\mathcal{Y}|$ vectors for output predictions over all classes in $\mathcal{Y}$. The generic model is trained on the large dataset, and we want to personalize it into a few-way classification task $k$. One effective method is to build a personalized classifier $\mathbf{w}_k$ by selecting only the row vectors in $\mathbf{w}$ for the relevant classes. Therefore, the personalized model for task $k$ are $\theta_k = \{\psi, \mathbf{w}_k\}$, and this approach is called classifier selection, which serves as a strong baseline [13].

**TAPER.** We briefly introduce TAPER [13] proposed by the original paper on train-once-for-all personalization and discuss its limitations. The main idea of TAPER is to train several experts (bases) and learn a mixture network to fuse these experts into a personalized model. It has three stages as follows.

- **Stage 1:** train a generic model on the large dataset.

- **Stage 2:** divide the dataset into several shards and finetune the generic model on each shard respectively for specification. Each finetuned model can be seen as a domain expert.

- **Stage 3:** For a given personalized task, learn an MLP mixer (i.e., the generator $G$) whose input is the text embedding of the task description and the output is the aggregation weights of the expert models. Then, weighted aggregation is conducted to merge several expert models into a personalized one. Also, the expert models can be finetuned during personalization.

TAPER requires finetuning the expert models on the target task, so it is not applicable to unseen tasks without having task-specific data. Also, the MLP mixer only generates the aggregation weights instead of the parameters, so it has limited generalization and expressiveness. While in our design of `Tina`, we try to construct an end-to-end text-to-model system that can understand the hyper-knowledge residing parameters and can generalize to unseen tasks, even unseen classes.

### 2.1.3 Dataset Preparation and Description

We introduce how to conduct datasets for training `Tina` and elaborate on the differences in training and inference between `Tina` and TAPER.

**Training data preparation for `Tina`.** `Tina` takes the personalized model parameters as training data for diffusion training, and the dataset is conducted in two stages. i) Stage 1: Similar to TAPER, we train a generic model on the large dataset to let the model have a generic capability on all classes. ii) Stage 2: We craft the personalized tasks and finetune the generic model on the personalized tasks to obtain the personalized models (p-Models) for `Tina` training. For each personalized task $k$, we select the corresponding $|\mathcal{Y}_k|$ classes out of $|\mathcal{Y}|$ classes to craft the data for p-Model, and then finetune to get a p-Model as a data sample for `Tina`. Each data sample for `Tina` contains the "(task description, p-Model)" pair.

**Testing data preparation.** The overall demonstration of data partitions can be found in Figure 2. The blue blocks refer to the training data, and the green blocks are the testing data. For testing, there are two kinds of evaluation metrics: **i)** In-distribution (ID, the light green blocks): the personalized tasks are seen during training of the generative model $G$, and $G$ generates the p-Models tested on the testset of each seen task. **ii)** Out-of-distribution (OOD, the dark green blocks): the tasks are unseen during the generator $G$'s training, and $G$ directly generates the p-Models from the task prompts (the text descriptions). We note that the original TAPER cannot be tested on the OOD tasks since it requires the target personalized training data for finetuning the expert models. To remedy this, we derive TAPER-Mixer to only train the mixer without finetuning the experts and verify its OOD generalization on unseen tasks.

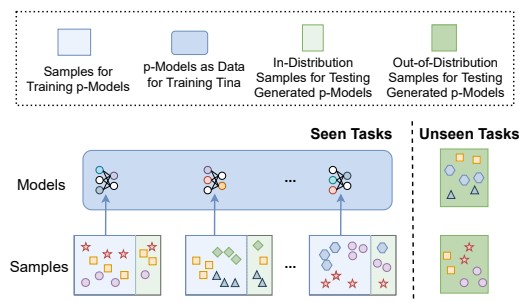

Figure 2: **Description of the training and testing data for `Tina`.** p-Model is short for personalized models. The blue blocks are for training, and the green blocks are for testing.

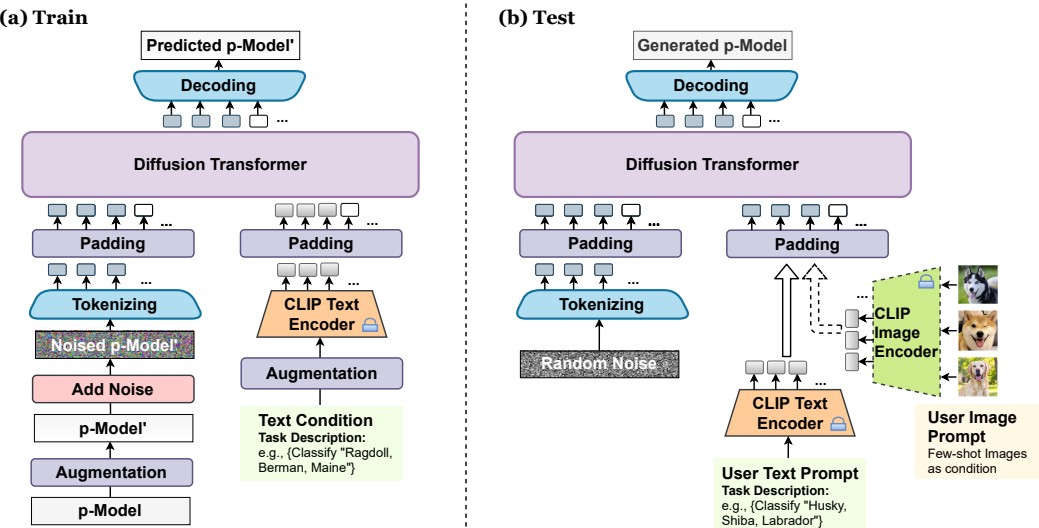

Figure 3: **Framework overview of** Tina.

## 2.2 Proposed Tina: Text-conditioned Neural Network Diffusion Model

### 2.2.1 Framework Overview

We present Tina, a text-conditioned neural network diffusion model for train-once-for-all personalization. The framework overview is in Figure 3. Generally, Tina consists of DiT and CLIP encoders for generating personalized models from text prompts. During training, we use the CLIP text encoder for encoding texts, and due to the alignment of image and text in CLIP, during inference, Tina can also take images as prompts by utilizing the CLIP image encoder. Additionally, we devise an effective data augmentation approach to enable training Tina under limited samples. We also propose a classification sequence padding strategy to enable Tina can generate models with different lengths of classes for further personalization.

### 2.2.2 Architecture and Training Objective

We use diffusion models as the generative model and follow the main architecture of G.pt [16] that uses a diffusion transformer as the backbone. Analogous to the optimization process that takes random initialization as inputs and outputs the trained models, the diffusion process takes the noise as inputs and gradually denoises to recover the original distributions. Previous works have shown the rationale of neural network diffusion [16, 15, 18]. We choose DiT as the backbone because it can be easily scaled up and is shown to have great generalization and expressiveness. We use signal prediction for the diffusion process and inherit the architecture of GPT-2 [8] as the transformer. The used text encoder is the pretrained ViT-B/32 in CLIP [17].

**Training objective.** Denote the training set of Tina as $\mathcal{K}$, where each piece of data is a (task description, p-Model) tuple, notated as $(t_k, \theta_k)$ for task $k \in \mathcal{K}$. We denote the CLIP text encoder as $T$, and given the task description $t_k$, the text embedding is $T(t_k)$. The text encoder is frozen during training.

---

**Algorithm 1** Tina Training

1: **Input:** Number of training iteration $N_{\text{iter}}$, p-Model dataset $\mathcal{K} = \{(t_k, \theta_k)\}_{k=1}^{K}$, Tina, diffusion process length $J$, diffusion cumulative variance schedule $\bar{\alpha}$.
2: **Initialize:** Learnable parameters $\phi$ for $G$
3: **for** $i = 1, 2, ..., N_{\text{iter}}$ **do**
4:      # Sample a mini-batch of data
5:      $(t_k, \theta_k) \sim \mathcal{K}$
6:      # Noise p-Model parameters
7:      $j \sim U(\{1, ..., J\})$
8:      $\theta_k^j \sim \mathcal{N}(\sqrt{\bar{\alpha}_j}\theta_k, (1 - \bar{\alpha}_j)I)$
9:      # Compute the predictions
10:     $\hat{\theta}_k \leftarrow G_\phi(T(t_k), \theta_k^j, j)$
11:     # Compute the loss
12:     loss $\leftarrow ||\hat{\theta}_k - \theta_k||_2^2$
13:     # Update DiT's parameters
14:     $\phi_{i+1} \leftarrow$ update(loss; $\phi_i$)
15: **end for**

---

Our DiT model $G_\phi$ takes two vectors as input: the text embedding $T(t_k)$ as conditions and the noised p-Model parameter vector $\theta_k^j$, where $j \in [J]$ denotes the timestep in the diffusion forward noising

process. The learning objective of diffusion is to minimize the simplified variational lower bound, which reduces to predicting the denoised p-Model parameters:

$$\min_{\phi} \mathcal{L}(\phi) = \sum_{k \in \mathcal{K}} \sum_{j \in \mathcal{J}} ||\theta_k - G_{\phi}(T(t_k), \theta_k^j, j)||_2^2, \tag{1}$$

where the timestep $j$ is embedded in DiT by frequency-based encoding [19]. The detailed training procedure is in Algorithm 1. We use DDPM sampling [9]; add Gaussian noise depicted by the $\bar{\alpha}$ to $\theta_k$ and gradually denoising it.

### 2.2.3  Design Details

We elaborate the design details of `Tina`.

**Parameter tokenization.** For a p-Model's parameters $\theta_k$, we first flatten all the parameters into a 1-D vector and chunk/tokenize the parameters within each layer. If the chunk size is $M$ and the number of parameters in a certain layer is $N$, so for this layer, there will be $ceil(N/M)$ tokens. For some layers smaller than $M$, the whole layer is a token.

**Text embedding.** Assume the personalized task is a classification task that has $c = |\mathcal{Y}_k|$ classes. The task description $t_k$ is an ordered list of the classes' text descriptions, of which the simplest form is the class entity, e.g., "telephone" and "rabbit". The generated p-Model is expected to have the correct predictions in the same order with $t_k$. In other words, we need `Tina` to learn the correct classifier orders as the text prompts, which is sequence-to-sequence modeling. Therefore, unlike TAPER, which averages the class embeddings into one, we make every class description as a token by CLIP text encoder and concatenate them in order with position encoding.

**Encoding and decoding of tokens.** We use linear layers as encoders for mapping the parameter tokens and text embedding tokens to the hidden size of DiT. Each token has a different linear layer without weight sharing. The decoders are similar to encoders, which use linear layers, and the encoders transform the transformer's hidden size back to the p-Model's parameter dimension. Between the encoders and decoders, there are transformer attention layers akin to GPT-2.

**Data augmentation.** In [16], the permutation invariance property [20, 21, 22] is utilized for data augmentation by randomly permuting the neurons without changing the function. However, in our scenario, we find this augmentation will even impede training. We hypothesize that the personalized models are finetuned from the same generic model, so they may lie in the same or close loss landscape basins; as a result, permutation augmentation will disturb network representations and impair `Tina` training. Further, we develop an effective *classifier augmentation* strategy to speed up `Tina` training under limited data by randomly permuting the order of classes in the task description and also the order of corresponding classifier vectors during training. This data augmentation improves sample diversity and helps the DiT better learn the description-to-classifier sequence modeling in a position-aware manner.

**Parameter inheritance.** In [16], the authors release a pretrained checkpoint of G.pt, which is also DiT for parameter generation. G.pt is pretrained on large datasets of optimization checkpoints; though it has different conditions, designs, and scenarios from ours, we explore whether we can inherit some parameters from the pretrained checkpoints to speed up and boost training. Considering the model sizes and architectures are different, we use a strategy similar to bert2BERT [23, 24, 25] for inheriting parameters.

**Classification sequence padding.** We study how to incorporate more personalized settings where diverse users request for tasks with different numbers of classes. In language models [26, 5], padding is used to enable sequence-to-sequence learning with different input and output lengths. Inspired by this, we use the padding technique to enable the description-to-classifier sequence of different classification lengths. Specifically, if the user's number of classes is smaller than the maximal length, we pad missing classes with tokens '`<->`' in the task description list and mask the corresponding classifier vectors with zero-like tensors. We denote this strategy as *classification sequence padding*, and `Tina` can learn to adapt to any number of classes within the maximal length.

## 3  Experiments

### 3.1  Experimental Setups

**Datasets and p-Models.** We use three datasets to conduct experiments: Mini-ImageNet [27, 28], CIFAR-100 [14], and Caltech-101 [29]. Mini-ImageNet is a subset of the ImageNet dataset, primarily

Table 1: **Main results across different datasets and models.** The best results are in **bold**.

| Dataset | Mini-ImageNet | | CIFAR-100 | | Caltech-101 | | **Avg** | |
|---|---|---|---|---|---|---|---|---|
| p-Models. | CNN | ResNet | CNN | ResNet | CNN | ResNet | CNN | ResNet |
| In-distribution Personalization | | | | | | | | |
| Generic Model | 19.76 | 39.32 | 28.72 | 51.24 | 29.14 | 47.95 | 25.87 | 46.17 |
| Classifier Selection | 51.74 | 71.49 | 64.83 | 84.01 | 56.07 | 74.75 | 57.55 | 76.75 |
| TAPER-Mixer | 52.16 | 65.50 | 67.71 | 75.12 | 58.48 | 77.92 | 59.45 | 72.85 |
| Tina | **54.08** | **74.99** | **68.35** | **86.46** | **58.69** | **78.36** | **60.37** | **79.94** |
| Out-of-distribution Personalization | | | | | | | | |
| Generic Model | 18.55 | 39.80 | 29.88 | 52.24 | 29.14 | 50.56 | 25.86 | 47.53 |
| Classifier Selection | 51.02 | 72.47 | 64.15 | 83.94 | 56.44 | 76.03 | 57.20 | 77.48 |
| TAPER-Mixer | 51.64 | 67.03 | 66.85 | 72.30 | 58.93 | 79.65 | 59.14 | 72.99 |
| Tina | **53.31** | **75.34** | **67.14** | **86.63** | **59.27** | **79.69** | **59.91** | **80.55** |

used for few-shot learning tasks. CIFAR-100 is a popular benchmark dataset for image classification tasks. Each class contains 600 images, divided evenly into 20 superclasses and 100 classes. Caltech-101: A dataset for object recognition featuring diverse images with varied resolutions and quality. It includes 101 categories, each containing 40 to 800 images, offering a wide range of objects and scenes compared to CIFAR-100 and Mini-ImageNet. For the images with different resolutions, we resize them into $32 \times 32$ for unified modeling. The personalized tasks are crafted by selecting 10 classes out of the 100/101 total classes. If not mentioned otherwise, the number of p-Models (i.e., personalized tasks) for training Tina is 1000.

We use two architectures for personalized models: a simple CNN (dubbed as CNN) and ResNet-20 (dubbed as ResNet). The CNN architecture follows [16], which consists of 2 layers, and the number of parameters is approximately 5K. We take all the parameters of CNN as the input and output of Tina. But for ResNet-20, the number of parameters is nearly 272k, which is too large for Tina's generation. Thus, we explore partial parameter generation following [15]. We only personalize the classifier layers for parameter generation, nearly 640 parameters.

For more details about data preparation and p-Models, please refer to Appendix A in the appendix.

**Compared baselines.** We follow the baselines used in the original paper of train-once-for-all personalization [13]. As described in subsection 2.1.3, we use the generic model trained in stage 1 as a baseline, showing the performance without any personalization. Further, we compare the classifier selection method described in subsection 2.1.2, which serves as a strong baseline for personalization [13]. The vanilla TAPER [13] requires finetuning the expert models on the target tasks and cannot generalize on out-of-distribution personalization where only target text descriptions are available. For fair comparisons, we adopt TAPER-Mixer, which adopts the mixer of TAPER for generating the aggregation weights, and the MLP-based mixer can generalize on unseen tasks.

**Evaluation metrics.** For Table 1, we compare in-distribution personalization and out-of-distribution personalization as elaborated in subsection 2.1.3. For other tables and figures, we report the out-of-distribution personalization as p-Acc.

**Hyperparameters.** The detailed hyperparameters can be found in subsection A.5 in the appendix.

## 3.2 Results under Different Datasets

In Table 1, we evaluate the performance of our proposed method, Tina, against several baseline methods including Generic Model, Classifier Selection, and TAPER-Mixer across various datasets and model architectures for the task of train-once-for-all personalization. It is found that the Generic Model has inadequate performance, validating the need for personalization techniques. For the personalization methods, the results demonstrate that Tina consistently outperforms all baseline methods across both in-distribution and out-of-distribution personalization scenarios. Though Tina is a text-to-model foundation model, it is worth noting that Tina shows intelligence of personalization under limited data (nearly 1000 samples). Specifically, for in-distribution personalization, Tina achieves significant improvements with an average score of 79.94, surpassing the next best method, Classifier Selection, by a margin of 3.19. Similarly, for out-of-distribution personalization, Tina leads with an average score of 80.55, which is a notable increase over the second-best performing method by 2.78. It is notable that TAPER-Mixer shows performance gains over Classifier Selection in CNN

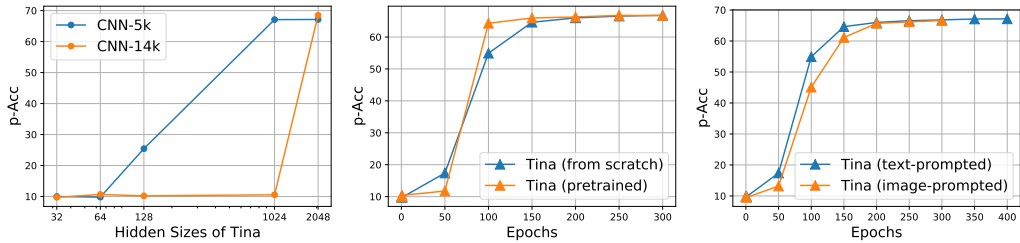

(a) **Scaling the parameters of DiT.**   (b) **Parameter inheritance.**   (c) **Training images as prompts.**

Figure 4: `Tina` **capability analysis w.r.t. different parameterization and training schemes. (a) Scaling the parameters of DiT in** `Tina`**.** CNN-5K (14K) means the p-Model is a CNN with 5K (14K) parameters. From 152M (hidden size 32) to 789M (hidden size 2048), scaling helps in the emergence of intelligence. **(b) Parameter inheritance from pretrained G.pt** helps speed up training in the early. **(c) Training** `Tina` **with image-prompted data versus text-prompted data.** The text-prompted has faster convergence.

but has marginal results in ResNet. Also, TAPER-Mixer has inferior performance compared with `Tina`, showing the advantages of `Tina` as a generative model in parameter generation. TAPER-Mixer only *learns to merge* the expert models, while `Tina` *learns to directly generate* the parameters.

### 3.3   In-depth Analysis of `Tina`

`Tina` shows great potential for text-to-model generation for personalization. We have made several in-depth analyses to better understand the capabilities and boundaries of `Tina`, and we will show insights into how `Tina` learns hyper-level world knowledge as well as its limitations for future research. If not mentioned otherwise, we use CIFAR-100 as the dataset for analyses.

**Scaling studies for** `Tina`**.** Scaling law was found for transformer-based foundation models that scaling the parameters, data, computes can bring intelligence emergence. In Figure 4 (a), we scale the parameters of `Tina` by changing the hidden sizes ranging from 32 (152M parameters) to 2048 (789M), and we test two sizes of p-Model. It is found that when `Tina` is small, it fails to generalize, especially when the p-Model has a higher parameter dimension. The intelligence emerges when scaling `Tina` at large sizes (e.g., 1024 or 2048 hidden sizes), but the scaling effect is saturated if reaching the upper bound performance of personalization. We also scale the input, also the generated, dimensions (i.e., p-Model sizes) and the training data in Figure 5. It is found that a larger input dimension is harder to learn and requires larger sizes of training data to converge and generalize. The generalization of `Tina`

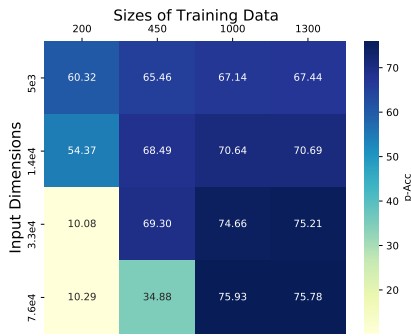

Figure 5: **Scaling the input dimensions and training data for** `Tina`**.**

can benefit from larger training data, but it has diminishing marginal returns. Generally, larger p-Models, larger training samples, and larger model sizes make `Tina` reach higher p-Acc, and it demonstrates the increasing expressive power of `Tina` by scaling, which is consistent with previous DiT works [12, 16, 3]. The scaling property indicates the great potential of `Tina` for more complex and challenging text-to-model scenarios.

**Parameter inheritance.** We verify whether `Tina` can benefit from pretrained parameters. We inherit the parameters from G.pt's [16] checkpoints by the bert2BERT-like method [24]. From Figure 4 (b), it is found that parameter inheritance from pretrained models can help `Tina` to converge faster, but the final p-Accs are similar.

**Training images as prompts.** In the original design of `Tina`, the texts are used for the prompts encoded by the CLIP text encoder. We train a `Tina` with image prompts using CLIP image encoder, and the results are in Figure 4 (c). For each class, we randomly select one single image as the prompts. It is found that text-prompted `Tina` converges faster than the image-prompted, though the final p-Accs are similar. This is intuitive to understand since texts are known to have higher knowledge density than images [30, 17], that the class text has richer knowledge representations than a single image.

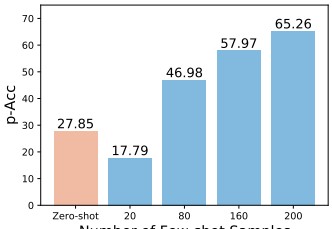
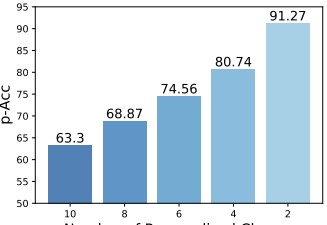
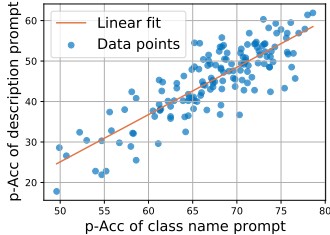

| (a) **Image as prompts.** | (b) **Different personalized classes.** | (c) **Descriptions as prompts.** |

Figure 6: `Tina` **capability analysis w.r.t. different prompt schemes. (a)** Train text-prompted `Tina` and verify **the zero-shot and few-shot abilities of using images as prompts. (b) The accuracies of p-Models generated by `Tina` vary with different numbers of classes.** Classification sequence padding is used, and the maximal sequence length is 10. **(c)** Train class-name-conditioned `Tina` and verify its **zero-shot ability on the natural language descriptions generated by GPT-4.**

Table 2: **Zero-shot transfer of `Tina` to unseen classes.** We test the generalization capability of `Tina` to unseen classes that have similar textual similarity with the seen ones.

| Settings | 0% unseen tasks | 20% unseen tasks | 40% unseen tasks | 60% unseen tasks | 100% unseen tasks |
|---|---|---|---|---|---|
| TAPER-Mixer | 60.27 | 51.94 | 42.48 | 31.45 | 0.0 |
| Tina | **62.51** | **55.36** | **49.17** | **42.78** | **30.93** |

**Testing images as prompts.** We train text-prompted `Tina` and verify its zero-shot and few-shot abilities on image prompts, and the results are in Figure 6 (a). Due to the alignment of texts and images in CLIP, `Tina` shows zero-shot ability on image prompts. By few-shot finetuning on image prompts, `Tina` can reach comparable performances to the text-prompted model. We note that the image-prompted ability is important in practical personalization scenarios, because some users may have few images and want a personalized model for those. The images are too few to train a model from scratch, but thanks to the generative power of `Tina`, we can generate a p-Model given image prompts by utilizing `Tina`'s vision-language-parameter-aligned knowledge.

**Varying the number of personalized classes.** Without changing architecture, `Tina` can adapt to any personalized classes within the maximal supported length due to the padding design. In Figure 6 (b), we test the p-Models with different numbers of classes, generated by one `Tina`. The maximal classification length is 10. It is shown that the generated p-Models reach higher p-Accs when the number of classes is fewer, which is consistent with common sense that fewer classes are easier to personalize.

**How `Tina` understands world knowledge I: natural language descriptions as prompts.** In our implementation of `Tina`, we adopt a simple prompting that uses the class names as the text prompts. We verify whether `Tina` actually learns the knowledge in the case where the prompts are replaced by the natural language descriptions at test time. We generate the language descriptions of classes with the assistance of GPT-4 [31], and we make sure that the descriptions do not include the original class entities. The exemplars are in Table 4 of the appendix. From Figure 6 (c), the results reveal that `Tina` has zero-shot generalization ability when the prompts are unseen language descriptions, though the p-Accs are lower than the ones of the class-named prompts. It shows that `Tina` is not just memorizing the class names but also generalizing and understanding the knowledge behind the names and the nuances inherent in the text semantics.

**How `Tina` understands world knowledge II: generalization to unseen classes/entities.** We divide the CIFAR-100 dataset into two disjoint shards of classes and train a `Tina` on one shard, then verify its generalization on the unseen classes of another shard. Results in Table 2 showcase that `Tina` has the intelligence to generalize on unseen classes, while TAPER-Mixer fails when meeting 100% unseen classes. As a generative model, `Tina` can understand the hyper-level world knowledge embedded in model parameters as well as text semantics and generate models for predicting unseen entities.

### 3.4 Ablation of Design Choices of `Tina`

We make an ablation study for different design choices of `Tina`. The ablated designs are the ones different from previous literature, such as our design of classifier augmentation, G.pt's design of permutation augmentation [16], and TAPER's design of merge text embedding as one [13]. The results are in Table 3. Our classifier augmentation can boost the performance even under small training datasets.

Permutation augmentation has negative effects on generating personalized models, and we hypothesize that for `Tina`'s training data, the p-Models finetuned from the same generic model are located in a common loss basin, where permutations will disturb the shared representations.

Table 3: **Ablation study for different design choices of `Tina`.**

| Designs/Datasets | Mini-Imagenet | CIFAR-100 | Caltech-101 | Avg. |
|---|---|---|---|---|
| w/o classifier aug. | 32.45 | 49.61 | 41.61 | 41.22 |
| w/ permutation aug. | 9.88 | 10.14 | 10.59 | 10.20 |
| merge text embed. as one | 10.04 | 10.35 | 10.78 | 10.39 |
| Tina (completed) | **53.31** | **67.14** | **59.27** | **59.91** |

In addition, merging the text embeddings into one will hinder the DiT from learning the sequential classifications, making `Tina` bad in generalization.

## 4   Related Works

**Diffusion models.** Originating from non-equilibrium thermodynamics [32, 33], diffusion models have evolved significantly. DDPM and DDIM pioneered forward-and-reverse processes in text-to-image generation [9, 34]. Guided-based diffusion models [35] surpassed GAN-based methods in image generation quality. Subsequent models like GLIDE [36], Imagen [37], DALL·E 2 [2], and stable diffusion [4] further advanced image generation and art creation. The diffusion transformer (DiT) [12] introduced a scaling law, with OpenAI's Sora [3] being a notable application in text-to-video generation, employing DiT architecture at a billion-scale.

**Parameter generation.** Learning to optimize explores neural networks learning update rules for others [38, 39, 40, 41]. Hypernetwork [42] is a meta learning approach that uses networks to modify neural network parameters, differing from our approach of mapping language space directly to parameter space. Hypernetworks are used in federated learning [43], few-shot learning [44], and model editing [45]. A concurrent work ModelGPT [46] customizes models by large language models and hypernetworks, while `Tina` uses conditional neural network diffusion for a different task—train-once-for-all personalization. Neural network diffusion [16, 15] is recently proposed to mimic optimization rules via diffusion for parameter generation, but previous works haven't explored sufficient use cases of such techniques.

For more detailed related works (e.g., the works about personalization), please refer to Appendix B.

## 5   Discussions

**Limitations.** Despite the merits of `Tina`, it has some current limitations. One bottleneck is the input dimension; due to our computation limits, `Tina` currently supports lightweight models as inputs, and it requires huge computation resources to fully generate large models with millions of parameters. On the one hand, a larger input dimension needs exponentially larger `Tina` parameters, so more GPUs. On the other hand, a larger input dimension needs more data to converge or generalize, requiring more compute hours. As a remedy, we tried to train a variational autoencoder (VAE) for encoding the p-Model parameters into a low-dimension latent space as in [15], but the VAE cannot generalize, suggesting more advanced techniques are needed. Another limitation is the generality of `Tina`, that one single `Tina` cannot generate personalized models across different sizes and different modalities; in the future, large-scaling pretraining for `Tina` may be promising to reach this goal.

**Broader impacts.** `Tina` is the preliminary work of text-to-model generation and will have broader impacts on the machine learning community, especially in the field of generative AI and model personalization. Though in this initial version of `Tina`, we only showcase its great potential in image classification tasks, `Tina` is prospective in a wide range of applications and tasks, such as natural language processing, audio recognition, and recommender system. Also, `Tina` has opened more potential directions for neural network diffusion, and we believe it can inspire more interesting works in the future.

## 6   Conclusion

In this paper, we present `Tina`, a text-to-model neural network diffusion model for train-once-for-all personalization. `Tina` has shown its great capability in generating personalized models from text prompts, and it can generalize to in-distribution as well as out-of-distribution tasks, zero-shot/few-shot image prompts, natural language prompts, and unseen classes. `Tina` also supports personalization under different numbers of classes. This paper explores the potential of text-to-model generative AI and opens new applications for neural network diffusion in end-user personalization.

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

# Appendix

## A  Implementation Details

### A.1  Dataset Preparation

**Mini-ImageNet.**  The Mini-ImageNet dataset [28] is a sub-dataset of ImageNet [27], which is widely used in few-shot learning. It selects 100 categories from ImageNet1K. The trainset contains 600 labeled images for each category, a total 60,000 images, and the testset contains 100 labeled images for each category, a total of 10,000 pieces.

**CIFAR-100.**  Each image in CIFAR-100 [14] has two labels: superclass and subclass. There are 500 training images and 100 testing images per subclass. CIFAR-100 has 20 superclasses, and each superclass has 5 subclasses.

**Caltech-101.**  Caltech-101 [29] is an objects image dataset with 101 categories. Approximately 40 to 800 images per category, most categories have around 50 images, 8677 images in total. We divide it into a trainset and a testset according to the ratio of 8:2.

When creating the p-Model datasets, we strive to maintain a consistent frequency of occurrences for each class, while simultaneously varying the combinations of different classes in various orders. For each dataset, we randomly permute the order of all classes, divide them into ten classes, and train on the respective classes to construct p-Models. This approach allows us to generate 10 distinct class models for each dataset. We utilize various random seeds to control the generation of class combinations, ensuring we acquire sufficient p-Models. We randomly selected 150 data from the original training data as the out-of-distribution testset.

For CIFAR-100, it has two classification methods: superclass and subclass. In order to increase the diversity and semantics of p-Model data, we use a more complex way to set up the classes included in each model. (1) The classes trained by each model come from different superclasses. This ensures a wide range of semantic variations. (2) Part of the classes trained by each model come from the same superclass. The selection of these classes is done randomly. (3) The classes trained by each model only come from two different superclasses. In the trainset and testset, we distribute these three division methods in quantity according to 3:2:1.

### A.2  Example of class description from GPT-4

For the word of each class, we use GPT-4 to provide a more detailed and standardized description and definition. Some examples are shown in Table 4.

Table 4: **Natural language descriptions of the class names from GPT4.**

| class | description of the class from GPT4 |
|---|---|
| "boy" | "a male child or young man" |
| "girl" | "a female child or young woman" |
| "apple" | "a round fruit with red, green, or yellow skin and a crisp, sweet flesh" |
| "pear" | "a sweet, juicy fruit with a thin skin and a rounded base tapering to a stalk" |
| "orange" | "a round, juicy citrus fruit with a tough, bright orange rind" |

### A.3  Data Preparation for Experiments of Unseen Classes

We divide the 100 classes in CIFAR-100 evenly into two groups/shards. The classes belonging to one group serve as the training model data, while the classes in the other group are intentionally excluded from appearing during the training process. When making these divisions, we take care to distribute categories with similar characteristics into separate groups. For instance, we separate the apple and the orange, both being common fruits, into different groups. Similarly, the bear and the lion, both large carnivorous mammals, are divided, and the boy and the man, both representing the male gender, are also separated accordingly.

### A.4  Detailed Implementations of Methods

We first train the model on the entire dataset for 50 epochs to obtain a stage-one model.

**Classifier Selection:** Based on the stage-one model, for each classification task, we only retain the vector representing the corresponding class on the classifier and set the vectors for all other classes to zero.

**TAPER-Mixer:** We set up two base models and split the dataset into two shards based on the classification labels. Each base model is initialized using the parameters of the stage-one model and fine-tuned on one of the sharded datasets for 5 epochs. In stage 3, we use the class order of the p-Model in the trainset to train the mixer for 5 epochs, and during the testing phase, the mixer remains frozen.

`Tina`: For each p-Model data, we initialize it using the parameters of the stage-one generic model as a starting point. At the same time, each class is sequentially reorganized as labels ranging from 0 to 9 for training. We fine-tune the generic model for 10 epochs to obtain the p-Models. For ResNet-20, we only fine-tune the parameters of the classifier, while keeping the remaining network parameters frozen.

### A.5 Hyperparameters

In all experiments, we use the same hyperparameters for training. For the model structure, we set the hidden size to 2048, and the number of the encoder and decoder is 1. Each encoder and decoder has 12 layers, and each self-attention layer has 16 attention heads. For the training process, we divide the model parameters into chunks by layer, and the size of each chunk is 576. We set batch size 64, learning rate $4e^{-4}$, and the gradient clipping coefficient to 0.1.

### A.6 Environments and Resources

All our experiments are conducted on CPU Intel(R) Xeon(R) Silver 4210 CPU @ 2.20GHZ. We employ two Quadro RTX 8000 for data-parallel distributed training. When `Tina` generates a CNN neural network with 5,000 parameters, each GPU requires 20,000MB of memory, and training for 300 epochs takes approximately 5 hours.

## B  Detailed Related Works

**Diffusion models**  The origin of diffusion models is the study of non-equilibrium thermodynamics [32, 33]. In recent years, DDPM [9] and DDIM [34] have refined diffusion models to a higher level by transforming the paradigm into forward-and-reverse processes in text-to-image generation. Later on, guided-based diffusion models [35] found a better architecture to improve the image generation quality that could beat the GAN-based methods [47, 48]. Then, GLIDE [36], Imagen [37], DALL·E 2 [2], and stable diffusion [4] emerged and flourished in the field of image generation and art creation. In the work of diffusion transformer (DiT) [12], the authors found that if the basic architecture of diffusion models is changed to transformers, the scaling law emerges, that scaling the number of parameters can reach the increasing quality of image generation. Based on DiT, in Feb 2024, OpenAI launched Sora [3], a text-to-video model that can understand and simulate the physical world in motion. In Sora, the DiT architecture is used and scaled to the billions level.

**Parameter generation**  The field of learning to optimize studies how one neural network can learn the update rules (gradients) for optimizing another network [38, 39, 40, 41]. Besides, the studies of hypernetworks [42] focus on how to directly output or modify neural networks' parameters by a hypernetwork. Hypernetworks usually take models' parameters as input and generate parameters [43, 45], which is different from our paper, which directly maps language space into the parameter space. Hypernetworks were used to generate local models for federated learning [43], edge-cloud collaboration, few-shot learning [44], and model editing [45]. A concurrent work ModelGPT [46] also uses text prompts to generate customized models. However, ModelGPT didn't target the train-once-for-all personalization scenario, and it uses conventional hypernetwork and meta learning methods while our `Tina` adopts novel conditional neural network diffusion. Recently, empowered by the strong expressiveness of diffusion models, neural network diffusion [16, 15] was proposed to mimic the optimization rule by diffusion for generating the model parameters. The first paper is G.pt [16], which uses DiT to learn to generate the model given a targeted loss or accuracy, and it mimics the optimization process while achieving faster inference compared with vanilla optimization. However, G.pt may have limited use cases; it can only generate the models for the training tasks (i.e., the in-distribution in our paper's terminology), and the accuracies are upper-bounded by the accuracies of checkpoint models in the training datasets. p-diff [15] formally formulates the neural network diffusion problem and proposes to diffuse and generate the batch normalization layers for better accuracies, but the improvement may be marginal, and the diffusion design is not conditioned. It also

meets the dilemma of G.pt, which lacks a specific scenario and use case. Recently, GPD [18] uses the diffusion model for few-shot learning in smart city applications, which showcases the applications of neural network diffusion. However, GPD takes the smart city's knowledge graphs as prompts and is tailored for the specific smart city application that cannot be easily extended to other fields. Our `Tina` takes language texts as prompts, which is more flexible and can be extended to a wider range of applications for the personalization of user demands.

**Personalization**   Instead of training a generic model to provide many users with the same model service, personalization of deep learning models acknowledges users' characteristics and diversity and learns each a customized model. Personalization techniques were introduced in medical AI [49, 50, 51], recommendation systems [52, 53], large language models [54, 55], and especially federated learning [56, 57]. Personalized federated learning studies how to exploit the common knowledge of users and then use it to explore further personalization on users' local datasets under privacy constraints [56], and techniques like proximal descent [58, 57], network decoupling [56, 59], and clustering [60, 61] are used. Recently, the scenario of train-once-for-all personalization [13] was proposed to bridge the gap between edge-side and server-side personalization. Train-once-for-all personalization aims to utilize server-side computation and generic models for fast and effective personalized adaptation to meet the edge users' demands. The original method TAPER [13] finetunes the generic model into several base models and learns MLP-based hypernetworks as mixers to fuse the base models into the personalized one given users' task descriptions. However, the MLP mixer has limited generalization capability, and it cannot be applied to unseen tasks, whereas our `Tina` learns the text-to-model world knowledge and can be generalized to out-of-distribution samples, modalities, and domains.

