# OpenReview forum: "Text-to-Model: Text-Conditioned Neural Network Diffusion for Train-Once-for-All Personalization"
_NeurIPS.cc/2024/Conference — Submitted to NeurIPS 2024_

### Official Review · Reviewer_htFA · 2024-07-10

**Soundness:** 3
**Presentation:** 4
**Contribution:** 2
**Rating:** 4
**Confidence:** 4

**Summary:**

The paper introduces Tina, a text-conditioned neural network diffusion model designed for train-once-for-all personalization. Tina utilizes a diffusion transformer model conditioned on task descriptions embedded using a CLIP model. This innovative approach aims to generate personalized models for various end-users and tasks based on text prompts, demonstrating significant generalization capabilities even when trained on relatively small datasets (~1000 samples). The model is evaluated under zero-shot/few-shot image prompts, varying numbers of personalized classes, natural language descriptions, and predicting unseen entities to assess its understanding of world knowledge.

**Strengths:**

- The paper provides a comprehensive explanation of the design and framework of Tina.
- It conducts a detailed ablation study and experiments across different datasets.
- The topic is interesting, and the presentation is clear and easy to understand.
- very detailed and robust comparison with previous works.

**Weaknesses:**

- The model parameter size in the experiments is too small; larger models are needed to evaluate effectiveness.
- In Table 1, the results of direct fine-tuning should be included.
- We might need an ablation study on the impact of text prompts.
- We might need an ablation study to determine if the model merely memorizes and reproduces parameters.
- Figure 2 requires polishing for better clarity.

**Questions:**

Can you provide your prompts for Appendix A.2?
If the authors solve my questions, I will raise my scores.

**Limitations:**

The model size is too small in exp.

---

> ### Author Rebuttal · Authors · 2024-08-06
>
> # Response to Reviewer htFA
>
> Thanks for your valuable comments and kind attention. We appreciate the opportunity to address your concerns and give detailed responses as follows.
>
> > 1. Response to "The model parameter size in the experiments is too small; larger models are needed to evaluate effectiveness."
> >
>
> Thank you for this valuable comment. During the rebuttal, we added an experiment about using ViT as p-Model backbones to train Tina. The experimental results are shown in **Table B of the *Rebuttal PDF* in general response.** The results are promising that our Tina can reach 97.15’s accuracy in personalization when using the larger backbone ViT-B/32 pretrained by CLIP, and Tina also consistently outperforms the baselines.
>
> > 2. Response to "In Table 1, the results of direct fine-tuning should be included."
> >
>
> Thanks for the comment. We may note that direct finetuning is not applicable under out-of-distribution (OOD) tasks where only class names are given and training data are not accessible; as a result, we didn’t put direct finetuning in Table 1.
>
> However, in **Table A of the *Rebuttal PDF* in general response**, we provide some showcase comparisons between direct finetuning and Tina’s generated models. It is found that Tina has similar performances with direct finetuning on OOD tasks. It is notable that direct finetuning may serve as the performance upper bound of personalization because it assumes the accessibility to all data (both training and testing). However, in the setting of train-once-for-all personalization, the OOD ability is important for fast adaptation and personalization to end users only given the task descriptions.
>
> > 3. Response to "We might need an ablation study on the impact of text prompts."
> >
>
> In our initial submission, we made an analysis of text prompts by learning class name prompts and testing on natural language description prompts in Figure 6 (c).
>
> During the rebuttal, we have made a more in-depth ablation study on the impact of text prompts, as in **Table C of the *Rebuttal PDF*.** It is found that if training and testing use the same kind of text prompts, the performances are similar regardless of class-name prompting or description prompting. However, if the prompt strategies are different in training and testing, the results will degrade, and training in class name prompts has better transferability and generalization.
>
> > 4. Response to "We might need an ablation study to determine if the model merely memorizes and reproduces parameters."
> >
>
> Thanks for this insightful comment. Actually, we may suppose the experiments about Tina's generalization ability can prove and verify that our model is not merely memorizing but generalizing. Specifically, Table 1 (OOD personalization), Figure 6 (a) (train on text prompts and test on image prompts) and (c) (train on class-name text prompts and test on description prompts), and Table 2 (test on unseen entities/classes).
>
> During the rebuttal, we additionally make more validations, please refer to **Table A of the *Rebuttal PDF***.
>
> - **Euclidean Distances:** We have measured the Euclidean distances between model parameters. It is found that the generated models have obvious Euclidean distances from each other and also from the fine-tuned models.
> - **Ensemble Learning Ability:** Ensemble learning often demonstrates higher accuracy than individual models, which can be indicative of the diversity in the internal representations of different neural networks, meaning that the manifold representations of the model parameters are not identical. Therefore, we make the generated models and the fine-tuned one ensemble to see whether it benefits. The results show that the ensemble accuracies are higher than the averaged accuracy and even higher than the best individual accuracy.
> - Taking the above experimental results into consideration, it is evident that Tina is not merely memorizing parameters but generalizing.
>
> > 5. Response to "Figure 2 requires polishing for better clarity."
> >
>
> During the rebuttal, we have polished Figure 2 for better clarity, and the polished figure is shown in  **Figure A of the *Rebuttal PDF* in the general response**. We added more necessary legends in the figure and more detailed descriptions in the caption.
>
> > 6. Response to "Can you provide your prompts for Appendix A.2? "
> >
>
> The prompts for generating natural language descriptions of class names from GPT-4 are as follows.
>
> ~~~
> "I will give you a list containing various nouns. Please add some short, accurate, and common descriptions to these nouns that can accurately define these nouns, and then return to me a JSON file where the key is the name and the value is the corresponding description. An example of the description is: {"goblet": "a drinking glass with a base and stem", "anemones fish": "live associated with sea anemones", "chiffonier": "a tall elegant chest of drawers"}. The list to be processed is as follows:”
> ~~~
>
> The few-shot exemplars in the prompts are extracted from the WordNet definition.
>
> We will add the prompts in the appendix.
>
> > 7. Response to "The model size is too small in exp."
> >
>
> Please refer to the first response in your rebuttal thread and check the results of ViT in **Table B of the _Rebuttal PDF_**.
>
> ---
>
> [1] Garipov, Timur, et al. "Loss surfaces, mode connectivity, and fast ensembling of dnns." NeurIPS 2018.

---

> > ### Comment · Reviewer_htFA · 2024-08-13
> >
> > Thanks for your response and additional results.
> >
> > 1. I am still concerned about the model size. You can directly include model size parameter size in the Table and paper
> >
> > 2. The author might need to include statistical results in the Exp section, eg, Table 1, Table 2.
> >
> > 3. Please include the revised Figure and results in the final version.
> >
> > Overall, I will keep my score.

---

> > > ### Author Response · Authors · 2024-08-13
> > > **Thanks for your feedback**
> > >
> > > We thank the reviewer for the post-rebuttal feedback. Here, we provide the following response to address your remaining concerns.
> > >
> > > - We may kindly remind that due to NeurIPS’s policy, during rebuttal, we cannot edit and revise the submission (the paper PDF), hoping you could understand. The added results during the rebuttal are in the PDF of the [General Response](https://openreview.net/forum?id=nblJSwtdrJ&noteId=zVKd9OZqeO). We kindly promise that all the issues raised by the reviewers and all the results presented in the rebuttal will be included in the future version of the paper once accepted.
> > > - For the model size, actually, we have already included the details in the original submission, i.e., in Lines 240-245. Knowing your concern about the model size, during rebuttal, we have conducted experiments on ViT-B/32 (pretrained by CLIP), whose size is 78M (millions), much larger than CNN and ResNet, and the results are promising. We will take your advice on putting the details of model sizes in the Table captions of the paper.
> > > - For the statistical results, actually, the results in our paper (including the reviewer mentioned Tables 1 and 2) are statistical and fair since, for every setting, we test the personalization performances across over 100 tasks and take the average scores presented in the tables, and each task includes more than 1000 testing image samples. We believe this evaluation measurement is statistical, representative, and fair for the compared methods. Sorry for the confusion; we will include these evaluation details in the paper.
> > >
> > > We appreciate the opportunity to address your further concerns, and we sincerely apologize for any confusion and misunderstanding caused before. Your support is important to our paper and we deeply appreciate your valuable comments and feedback. If our response has relieved your concerns, we sincerely hope that you might consider increasing your score. This would be extremely important to us, and we hope for your understanding.
> > >
> > > Thanks again for your efforts and time in reviewing our paper.

---

### Official Review · Reviewer_Gsfd · 2024-07-13

**Soundness:** 2
**Presentation:** 3
**Contribution:** 3
**Rating:** 6
**Confidence:** 4

**Summary:**

To generate personalized models for a variety of end-users and tasks via text prompts, this paper introduces Tina, a text-conditioned neural network diffusion model. Tina employs a diffusion transformer model, complemented by a CLIP model to embed task descriptions. Remarkably, Tina demonstrates superior generalization capabilities even on small-scale datasets, performing well both within and outside the distribution of the training data. Furthermore, Tina exhibits robust performance under zero-shot/few-shot image prompts, natural language instructions, and unseen categories.

**Strengths:**

1.The method demonstrates excellent generalization, showcasing significant in-distribution and out-of-distribution performance even when trained on small datasets. It also exhibits robust behavior in predicting entities that have not been seen before.

2.Compared to existing text-to-image models (such as stable diffusion), text-to-video models (like Sora), and large language models (such as GPT-4), the concept of Tina, which generates personalized models suitable for specific tasks directly from text descriptions, is quite novel.

3.The experimental process is comprehensive and reliable. The paper conducts comparisons against baselines across multiple datasets, and it also undertakes experiments to validate generalization performance as well as performs ablation studies.

4.The experiments involves multiple datasets to verify the effectiveness of the proposed methods.

**Weaknesses:**

1.It is better to includes more comprehensive and competitive baselines to show the model’s effectiveness and advance. The two baselines come from one paper published in 2023. As for the experimental setting involves three widely-used datasets, I am wondering whether the experimental results excels or perform similarly to the SOTA performance on some of the three datasets. In other word, is it possible to apply the proposed strategy to some more advanced framework to make the performance similar to the SOTA, which ensure the proposed method have real applications in the real use.

2.The base model is CNN or ResNet in the experiments. Is the proposed method generalized to more advanced framework? Applying the proposed method on more advanced framework and obtain more advance performance indicates that the method has potential to be used in the real life.

3.We suggest providing necessary explanations in the captions of the model framework overview.

**Questions:**

None

**Limitations:**

The limitations are fine with me.

---

> ### Author Rebuttal · Authors · 2024-08-06
>
> # Response to Reviewer **Gsfd**
>
> Thanks for your valuable comments and kind attention. We appreciate the opportunity to address your concerns and give detailed responses as follows.
>
> > 1. Response to "I am wondering whether the experimental results excels or perform similarly to the SOTA performance on some of the three datasets. In other word, is it possible to apply the proposed strategy to some more advanced framework to make the performance similar to the SOTA, which ensure the proposed method have real applications in the real use."
> >
>
> Thank you for this valuable comment. During the rebuttal, we added an experiment about using ViT as p-Model backbones to train Tina. The experimental results are shown in **Table B of the *Rebuttal PDF* in general response.** The results are promising that our Tina can reach 97.15’s accuracy in personalization when using the SOTA backbone ViT-B/32 pretrained by CLIP, and Tina also consistently outperforms the baselines.
>
> > 2. Response to "The base model is CNN or ResNet in the experiments. Is the proposed method generalized to more advanced framework? Applying the proposed method on more advanced framework and obtain more advance performance indicates that the method has potential to be used in the real life."
> >
>
> Thank you for the comment. Please see the last response about Tina's performance on ViT.
>
> > 3. Response to "We suggest providing necessary explanations in the captions of the model framework overview."
> >
>
> Thanks for the kind suggestion. We will add explanations in the captions of Figure 3 (framework overview) in the next version. The explanations to add are as:
>
> >>
>
> **Framework overview of Tina.** **(a) Training stage.** The p-Models are firstly augmented by our classifier augmentation strategy and then noised according to the diffusion step. The p-Models are tokenized into chunks of vectors, and the classification sequence padding is optionally used if the classification length is shorter than the default. The CLIP text encoder is used to encode the users' text prompts during training. **(b) Testing stage.** Random noises are tokenized and denoised into parameters of p-Models. Thanks to the vision-language alignment of CLIP, Tina takes both text and visual prompts as the diffusion conditions.
>
> >>

---

> > ### Comment · Reviewer_Gsfd · 2024-08-10
> > **Thanks for responding my comments**
> >
> > I have checked the authors' rebuttal and reviewers' comments. The authors' rebuttal sounds fine with me, which addressed most of my concerns. The authors add some additional experiements (i.e. on ViT) for my concerns as well as other reviewers' questions.
> >
> > I still would like to vote for this paper. Considering other reviewers' comments and the overall quality of this paper, I tend to keep my score.

---

> > > ### Author Response · Authors · 2024-08-11
> > > **Thanks for the post-rebuttal response**
> > >
> > > Many thanks for your time and efforts in reviewing our work. We greatly appreciate your recognition of our work and are pleased to hear that we have addressed your concerns.

---

### Official Review · Reviewer_BHPs · 2024-07-14

**Soundness:** 3
**Presentation:** 3
**Contribution:** 3
**Rating:** 6
**Confidence:** 4

**Summary:**

This work introduces Tina, a text-conditioned neural network diffusion model designed for generating personalized models from text prompts. Tina aims to enable efficient personalization by training a generic model once and then customizing it for various end-user tasks using task descriptions. Leveraging a diffusion transformer model and CLIP-based text embeddings, Tina demonstrates the ability to generate models for a wide range of personalized tasks. The approach shows promising results in generalizing to both seen and unseen tasks, achieving state-of-the-art performance in several benchmarks.

**Strengths:**

1. Tina's train-once-for-all approach effectively addresses the need for personalized models without requiring extensive retraining, making it a practical solution for diverse end-user scenarios.
2. The model achieves competitive performance across multiple datasets, demonstrating its robustness and effectiveness in both in-distribution and out-of-distribution tasks.
3. Tina can handle various types of input prompts (text, images) and generalize to unseen classes and tasks, highlighting its versatility and potential for broader applications.

**Weaknesses:**

1. Some methodological details are sparse, such as the specific configurations and hyperparameters used for training Tina. Providing more granular details could help readers replicate the experiments.
2. The reason for adopting DiT as the weight generation model is not well justified. It would be good to see some results of adopting different kinds of diffusion models.

**Questions:**

1. It would be good to clarify how exactly the parameters are inherited from G.pt for initialization.
2. Tina encodes and decodes the parameter tokens with linear layers, which are trained along with the DiT model in an end-to-end manner. I'm curious about what are the benefits of this one-stage manner compared to the original neural network diffusion, which trains an autoencoder and diffusion model separately. And what would the performance be if you adopted the same pipeline of neural network diffusion for weight generation?

**Limitations:**

While limitations are discussed, the manuscript could benefit from a discussion of the scalability of Tina to larger datasets and more complex tasks.

---

> ### Author Rebuttal · Authors · 2024-08-06
>
> # Response to Reviewer BHPs
>
> Thanks for your valuable comments and kind attention. We appreciate the opportunity to address your concerns and give detailed responses as follows.
>
> > 1. Response to "Some methodological details are sparse, such as the specific configurations and hyperparameters used for training Tina. Providing more granular details could help readers replicate the experiments."
> >
>
> Thank you for the recommendation. We have included the main hyperparameters of training Tina in Appendix A.5. We will add more granular training details, such as the diffusion steps and the warmup schemes, in the future version. Also, to enable the effective reproduction of our results, we plan to release the source codes upon acceptance.
>
> > 2. Response to "It would be good to clarify how exactly the parameters are inherited from G.pt for initialization."
> >
>
> Specifically, we have adopted a similar method provided in the bert2BERT [1] to inherit the parameters in G.pt in a layer-by-layer manner (by the keys).
>
> - For the same key, when the shape of Tina is smaller than that of G.pt, we directly crop the parameters of G.pt into the shape of Tina for inheritance.
> - When the shape of Tina is larger than that of G.pt, we use the FPI (Function Preserving Initialization) strategy from bert2BERT to expand the dimensions. That is, for the parts of Tina that have extra dimensions, we first randomly select a column from G.pt for each column that needs to be expanded. The values of the selected column from G.pt and the corresponding expanded column are reassigned to be the values of each data point in the column divided by the respective number of columns.
> - When Tina has deeper layers than G.pt, we copy and stack the layers, which was also introduced in bert2BERT.
>
> We thank the reviewer for this valuable comment, and we will add these details in the appendix.
>
> > 3. Response to the rationale of using DiT and its advantages over other architectures.
> >
>
> Thanks for this valuable comment. Actually, we had a brief discussion on this point in the Discussions section (Lines 378-381) of the submission. We will give a more detailed discussion as follows.
>
> - The architecture of DiT has great expressive power in diffusion tasks, especially in text-to-image and text-to-video. Also, it has the good property of scaling law as studied in DiT’s original paper [2]; therefore, for such a difficult task as parameter generation, this property will facilitate the model to realize generalization by merely scaling up parameters (evidence in Figure 4 (a) of our paper).
> - The architecture of training an autoencoder and diffusion model separately is also worth studying, so we have already made some early attempts before submission and gained some insights.
>     - In our early attempts, we reproduced the architectures of p-diff in our text-to-model scenarios but found that the VAE is hard to converge and generalize if we generate a full model of CNN. This is also reasonable since in p-diff’s paper, it mainly generates the BN layers, which have few parameters, so the scalability of this architecture may be limited.
>     - We found that in the parameter generation task, the autoencoder is hard to train, and therefore, the latent space may be less representative; as a result, the diffusion process also has poor performance. In other words, in this architecture, the performances are bounded by the autoencoders and then the diffusion process, which are hard to improve and optimize.
>     - In addition, if within the same compute and parameter budget, our one-stage, end-to-end, and decoder-only-like architecture has higher utilization efficiency of the parameters and is more suitable for scaling.
> - However, we think the two-stage and autoencoder-based diffusion pipeline is also very meaningful and promising in weight generation. The key is how to learn an effective and representative latent space, and therefore, more advanced techniques are needed to explore its potential, which we may leave for future exploration.
>
> > 4. Response to “While limitations are discussed, the manuscript could benefit from a discussion of the scalability of Tina to larger datasets and more complex tasks.”
> >
>
> Thanks for the suggestion, and we will add this discussion to future versions of the manuscript. We give a brief discussion below.
>
> - In our paper, we have shown the preliminary results of the scaling law in Tina (Figure 4 (a) and Figure 5), and it is found that by scaling the parameters of DiT and the number of training data, Tina can generate and generalize on more complex models with higher dimensions. Therefore, Tina's scalability is promising and predictive once the computation resources are sufficient, which we leave for future explorations due to our current computation limits and budgets.
> - Additionally, during the rebuttal, we conducted experiments using more complex ViT as p-Models by generating personalized layers in **Table B of the *Rebuttal PDF* in general response**, and the results are also promising.
> - For more challenging tasks, such as full generation of ViT and training more enormous datasets, it is interesting for future works with sufficient computation budgets.
>
> ---
>
> [1] Chen, Cheng, et al. "bert2BERT: Towards Reusable Pretrained Language Models." ACL 2022.
>
> [2] Peebles, William, and Saining Xie. "Scalable diffusion models with transformers." ICCV 2023.
>
> [3] Wang K, Xu Z, Zhou Y, et al. Neural network diffusion[J]. arXiv preprint arXiv:2402.13144, 2024.

---

> > ### Comment · Reviewer_BHPs · 2024-08-11
> >
> > Thank you for the thorough and thoughtful rebuttal. The authors have successfully addressed my concerns, so I am increasing my rating from 5 to 6.

---

> > > ### Author Response · Authors · 2024-08-12
> > > **Thanks for the post-rebuttal response**
> > >
> > > Many thanks for your post-rebuttal feedback. We are deeply grateful for your efforts in engaging with our work and your support in raising your score. Thanks again for your time and attention.

---

### Author Rebuttal · Authors · 2024-08-06

# General Response

We thank the reviewers for their valuable comments and precious time.

We are deeply encouraged to receive recognition from the reviewers that the idea is *interesting and novel* (Reviewers Gsfd and htFA), the method is *practical* and *has excellent generalization and competitive performances* (Reviewers BHPs and Gsfd), the experiments are *comprehensive, robust,* *and reliable* (Reviewers Gsfd and htFA).

We find the reviewers' comments highly useful and will incorporate them into the future version of our paper.

Our responses to each reviewer are provided in each review thread. In addition, we put the revised figures and additional results in **the rebuttal PDF of this general response**, hoping it could relieve the reviewers’ concerns.

---

## Highlights of the Rebuttal PDF

- **Figure A: Polished version of Figure 2** [*Reviewer htFA*].
    - TLDR: We give a revised Figure 2, adding more figure legends and detailed descriptions in captions.
- **Table A: Evidence for Tina is not merely memorizing and reproducing parameters** [*Reviewer htFA*].
    - TLDR: We verify this point by comparing the ensemble learning accuracies and individual accuracies of generated models and measuring the Euclidean distances of model parameters.
- **Table B: Results of Tina, using ViT as p-Models** [*Reviewers BHPs, Gsfd, and htFA*].
    - TLDR: We conduct experiments by making ViT as the p-Model backbone. The results show that Tina reaches high performances (> 95%) and surpasses the baselines.
- **Table C: Ablation study of Tina on the impact of text prompts** [*Reviewer htFA*].
    - TLDR: We analyze the impacts of text prompting strategies in both training and testing.

---

### Decision · Program_Chairs · 2024-09-25

**Decision:**

Reject

**Comment:**

This paper suggested a new approach in learning model parameters for new tasks, by training a Diffusion model on text input. By describing the task in natural language, the model generates parameter weights. The current version of submission requires to address the following major concerns,

1- `scalability of experimentation`: The model parameter size in the experiments is too small; larger models are needed to evaluate effectiveness.

2- `statistical analysis`: lack of statistical results in the Exp section, eg, Table 1, Table 2.